# A Non-Isolated DC-DC Modular Multilevel Converter with Proposed Middle Cells

**Ming Huang**

Department of Electrical Engineering, School of Automation, Northwestern Polytechnical University, Xi'an 710054, China; minghuang@nwpu.edu.cn

**Abstract:** Unlike the modular multilevel converter (MMC) topology operated under the rectifier or inverter modes, control of the balanced state for the submodule (SM) capacitor voltage has emerged as the key issue for DC-DC MMCs. This is mainly caused by no balanced alternative powers being used for balancing SM capacitor voltages, which can be absorbed from the input or output DC sides of the converter. Typically, the alternative voltages and currents should be injected to achieve SM capacitor voltage balance in the DC-DC MMC. However, this solution is based on the cost of adopting the bulky LC filter components. For interconnecting different DC voltages in medium-voltage applications, this paper presents a non-isolated DC-DC MMC equipped with the proposed middle cells. It is intended to achieve DC voltage conversion without adopting bulky passive LC filters. On the one hand, the alternative currents, used for balancing the SM capacitor voltages, are arranged for flowing only within the phase legs of the proposed DC-DC MMC without disturbing the input current. On the other hand, through appropriate control of the middle cells, compensated components can be developed to eliminate the undesirable voltages on the output DC side. The middle cells of the proposed DC-DC MMC are supplied with the function of the active filter, which enables the DC-DC MMC system to escape the bulky LC components. Through theoretical analysis and a control strategy, the proposed DC-DC MMC has been analyzed comprehensively. Finally, the simulation and experimental results are verified to evaluate the effectiveness of the proposed DC-DC MMC.

**Keywords:** DC-DC modular multilevel converter (MMC); alternative voltage and current; middle cells

## 1. Introduction

In recent years, the increase in global power consumption introduced a huge challenge to the power grid, especially for the power transmission and distribution system. For high-voltage alternate current (HVAC) systems, the voltage conversion can be easily performed between the transmission and distribution systems by using power transformers, which are being developed based on proven manufacturing technologies. However, the bulk weight and size for the line frequency transformer are considered crucial problems, especially for transportation. Moreover, due to the rapid development of semiconductor technology since the 1970s, the HVAC system offers lower efficiency and greater investment costs compared with those of the high-voltage direct current (HVDC) system for long-distance power transmission [1]. Meanwhile, continuous inclusion of renewable sources (especially from wind farms) increases this tendency. Moreover, compared with the HVAC system, reactive power compensation equipment is no longer required in medium-voltage direct current (MVDC) and HVDC systems.

Thus, MVDC and HVDC systems have become promising schemes for power transmission. One of the application fields is offshore wind farms. However, different DC voltage levels are designed for these wind farms, which are mainly based on different transmission capacities. Therefore, for the requirements of the DC network interconnection

and power flow management, medium- and high-voltage DC conversion have shown significant demand, also serving as DC transformers. Recently, increased interest has been observed in the topologies of DC transformers [2–8]. The MMC, presented by Lesnicar and Marquardt R. [9], is considered the preferred topology for MVDC and HVDC systems. Provided by the modular feature of the MMC system, different DC voltage levels of power transmission can be achieved by simply cascading the corresponding amounts of the submodules (SMs). According to the reported literature [10–20], the DC-DC MMCs were evaluated for their potential implementation in medium transformers. Typically, the DC-DC MMC equipped with an AC transformer, which can be termed the isolated DC-DC MMC, has two MMC stages between each side of the medium-frequency transformer [10–13]. However, the problem is that each of the MMC stages suffers from a full power size, resulting in low efficiency and a higher cost. To cope with this issue, transformer-less MMC-based DC-DC converters were also investigated [14–20]. However, unlike the DC/AC or AC/DC MMC systems, alternative powers should be introduced for balancing the SM capacitor voltages of the non-isolated DC-DC MMC. Generally, a preferred way for providing balanced powers is to inject alternative voltages and currents into the phase legs. However, this solution introduces undesirable harmonics at the input and output sides of the DC-DC MMC. Generally, in order to suppress the influence of the inserted voltages and currents, the bulky passive LC filter components are essential.

To get rid of the bulk capacitor, the two-phase non-isolated DC-DC MMC topology demonstrates a promising feature for enabling alternative currents to flow only within the phase legs. However, the undesirable harmonics produced by the balanced alternative voltages appear at the output side of the DC-DC MMC. Typically, these harmonics can be suppressed by equipping a bulky and hard-designed coupled inductor [16]. To prevent the injected alternative current from invading the input DC current, a typical push–pull DC-DC MMC was presented [17]. However, bulky coupled inductors are still required to suppress the undesired harmonics caused by the injected alternative voltages at the output DC sides. A novel double-phase DC/DC MMC topology was proposed to prevent the inserted AC waveforms from disturbing both DC sides [18]. This was achieved by employing flying capacitors between the upper and lower arms of the proposed converter for establishing additional power routes to suppress the influence of the injected voltages on the output side. Additionally, a multistring one derived from the novel topology in [18] was also proposed for bipolar operation in high-power applications [19]. However, the high-voltage-based flying capacitors break the modular nature of the topologies. Based on this issue, a modified topology of [20] was also presented by replacing the flying capacitors with the cascaded SMs, which showed promising prospects for enlarging the applied voltage range. However, a complicated control algorithm and high cost accompany this kind of power converter. Researchers also investigated the polyphase DC-DC MMC [21]. In these converters, the sinusoidal alternative currents are controlled to flow within the three phase legs without disturbing the input current. However, the undesirable harmonics caused by the injected alternative voltages still need to be suppressed by equipping bulky reactors on the output side. The literature [22] proposed a stepped two-level control for the polyphase DC-DC MMC, which mainly focuses on producing a smaller capacitance and smoother circulating current. However, this solution introduces undesirable rush voltages on the arm reactors, resulting in higher requirements for the insulation and design of the reactor.

Of all the above-mentioned discussions, this paper presents a double-phase DC-DC MMC by adopting novel middle cells. As shown in Figure 1, it aims to achieve step-down DC voltage conversion without equipping the bulky passive LC filter components at the input and output sides of the DC-DC MMC. In the proposed DC-DC MMC, the alternative currents are controlled to flow within the double phases without disturbing the input DC current. Meanwhile, the influence of the alternative voltages on the output DC side is suppressed by the compensation components produced by the middle cells. Specifically, the middle cells in the proposed DC-DC MMC serve as the active power filter (APF), aiming to replace the position of the bulky passive LC filter with the active one. Additionally, due

to the specific circuit structure of the novel middle cell, the terminal voltages connecting the upper and lower arms are always controlled to be zero, which has no influence on the control of the DC bus voltage.

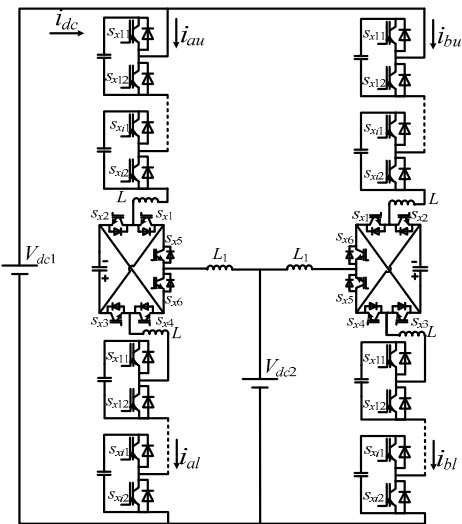

**Figure 1.** Basic structure of the proposed DC-DC MMC.

The rest of this paper is organized as follows. The basic topology structure and mathematical model of the proposed DC-DC MMC are presented in Section 2. In addition, the arm powers and SM capacitor voltages are also analyzed in this section. The control strategy of the proposed DC-DC MMC is discussed in Section 3. The simulation and the experimental results are presented in Sections 4 and 5 to verify the effectiveness of the theoretical analysis for the proposed DC-DC MMC. Finally, Section 6 concludes this paper.

## 2. Analysis of the Proposed DC-DC MMC

### 2.1. Basic Structure of the Proposed MMC Topology

Figure 1 illustrates the configuration of the proposed DC-DC MMC. It consists of the left and right phases. Each phase is composed of the upper arm, lower arm, and proposed middle cell. Both the upper and lower arms consist of $n$ cascaded HBSMs and the reactors, aiming to support the input DC bus voltage and produce the required alternative voltages. Emphatically, the reactors cascaded with the HBSMs are employed to suppress the inrush current and harmonics of the circulating current. Moreover, the additional inductors $L_1$ located at the output DC side are in charge of smoothing the output current and filtering the high-frequency harmonics. Note that the dominant harmonic voltages caused by the injected alternative voltages are mostly suppressed by the compensated middle cell voltages, which indicates that a smaller inductance for $L_1$ can be designed.

The structure of the HBSM in Figure 2a illustrates that each HBSM consists of two power switchers and one capacitor. During normal operation, the HBSM capacitor voltage is chopped by these power switchers with the complementary drive signals. The HBSMs, constituting the upper and lower arms of the proposed DC-DC MMC, are responsible for producing the required DC and AC voltage components. Typically, the AC components of the upper and lower arms in the proposed DC-DC MMC have the same amplitude with opposite signs, while the DC components of the upper and lower arms are determined by the defined input and output DC bus voltages. This setting of the solution enables the input DC bus voltage's maintenance. In particular, being similar to the MMC operated under the inverter and rectifier modes, the HBSM stacks in the proposed DC-DC MMC serve as the medium platform for power exchange between the input and output DC sides. Therefore, the fluctuated powers are inevitably imposed on the HBSMs and the proposed middle cells. In medium-voltage applications especially, these fluctuations significantly influence the distribution of the voltages and currents at the input and output DC sides. In

the proposed DC-DC MMC, the compensated components produced by the middle cells are adopted to suppress the influence of the fluctuated powers on the output DC side. In the meantime, the fluctuated powers are controlled to flow within the double phases only without disturbing the input DC side.

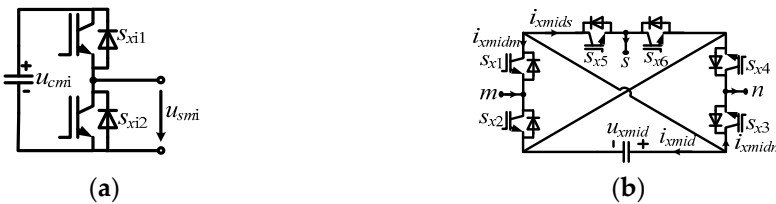

**Figure 2.** SM structure. (**a**) Half bridge-based SM. (**b**) Proposed middle cell.

As shown in the circuit structure depicted in Figure 2b, the middle cells in the proposed DC-DC MMC are placed for connecting the upper arm, lower arm, and the output DC side. It consists of six power switchers and one capacitor. The power switchers in the single middle cell can be divided into three pairs: $s_{x1}$-$s_{x2}$, $s_{x3}$-$s_{x4}$, and $s_{x5}$-$s_{x6}$. The power switchers in each pair of the middle cell are driven with complementary signals, which focus on producing the required middle cell terminal voltage by chopping the capacitor voltage. The available switching states of the middle cells in the proposed DC-DC MMC are listed in Table 1. Note that the middle cells can be regarded as the active power filters, supplying additional degrees to eliminate the undesirable voltages at the output DC side. According to the basic working principle of the middle cell, the switching functions in the middle cells can be expressed as

$$s_{xi} = \begin{cases} 1, & \text{when switch is on} \\ 0, & \text{when switch is off} \end{cases} \quad i = 1, 2, \ldots, 6 \tag{1}$$

where $s_{xi}$ is the switching state of the middle cell of phase $x$ ($x = a, b$). Therefore, based on the structure feature of the middle cell, the terminal voltages can be expressed as

$$\begin{cases} u_{xms} = (s_{x1} - s_{x5})u_{xmid} \\ u_{xsn} = (s_{x5} - s_{x3})u_{xmid} \\ u_{xmn} = (s_{x1} - s_{x3})u_{xmid} \end{cases} \tag{2}$$

where $u_{xms}$ is the terminal voltage between terminals $m$ and $s$ for the middle cell of phase $x$ ($x = a, b$), $u_{xsn}$ is the terminal voltage between terminals $s$ and $n$ for the middle cell of phase $x$ ($x = a, b$), $u_{xmn}$ is the terminal voltage between terminals $m$ and $n$ for the middle cell of phase $x$ ($x = a, b$), and $u_{xmid}$ is the middle cell capacitor voltage of phase $x$ ($x = a, b$).

**Table 1.** Switching states of the middle cell.

| Switching States | $u_{mx}$ | $u_{xn}$ | $u_{mn}$ |
|---|---|---|---|
| $S_{x1}, S_{x3}, S_{x6}$ are on; $S_{x2}, S_{x4}, S_{x5}$ are off | $u_{mid}$ | $-u_{mid}$ | 0 |
| $S_{x1}, S_{x3}, S_{x6}$ are off; $S_{x2}, S_{x4}, S_{x5}$ are on | $-u_{mid}$ | $u_{mid}$ | 0 |
| $S_{x1}, S_{x4}, S_{x5}$ are on; $S_{x2}, S_{x3}, S_{x6}$ are off | 0 | $u_{mid}$ | $u_{mid}$ |
| $S_{x1}, S_{x4}, S_{x5}$ are off; $S_{x2}, S_{x3}, S_{x6}$ are on | 0 | $-u_{mid}$ | $-u_{mid}$ |
| $S_{x1}, S_{x4}, S_{x6}$ are on; $S_{x2}, S_{x3}, S_{x5}$ are off | $u_{mid}$ | 0 | $u_{mid}$ |
| $S_{x1}, S_{x4}, S_{x6}$ are off; $S_{x2}, S_{x3}, S_{x5}$ are on | $-u_{mid}$ | 0 | $-u_{mid}$ |
| $S_{x1}, S_{x3}, S_{x5}$ are on; $S_{x2}, S_{x4}, S_{x6}$ are off | 0 | 0 | 0 |
| $S_{x1}, S_{x3}, S_{x5}$ are off; $S_{x2}, S_{x4}, S_{x6}$ are on | 0 | 0 | 0 |

### 2.2. Operating Principle of the Proposed Topology

For an MMC system operating under the inverter or rectifier modes, the upper and lower arm voltage references are specified, depending on the input and output rated voltages. Generally, the required alternative powers, which are used to balance the HBSM

capacitor voltages, are absorbed from the AC side of the MMC. However, unlike the inverter or rectifier MMC, since there are no alternative powers to be absorbed from the input and output sides of the DC-DC MMC, additional alternative voltages and currents should be injected in order to balance the SM capacitor voltages. Basically, the balanced state of the HBSM capacitor voltage of the proposed DC-DC MMC reflects that the average power of the single HBSM remains zero during one specific period, and this solution depends on the rearranged distribution of the injected alternative voltages and currents. On the one hand, in order to support the DC bus voltage, the injected alternative voltages are defined as the AC parts for the upper and lower arm voltage references. The difference lies in that the injected alternative voltages have the same amplitude with opposite signs for the upper and lower arms for each phase of the proposed DC-DC MMC. Meanwhile, as for the AC voltages of the upper arms (or lower arms) in the left and right phases of the proposed DC-DC MMC, the same amplitude with opposite signs for the injected alternative voltages should also be arranged. As for the undesirable voltages caused by the injected alternative voltages at the output DC side, the compensation components produced by the middle cells are employed to suppress the influence. On the other hand, to prevent the harmonics from invading the output side, the alternative current, adopted to cooperate with the alternative voltages for producing balanced powers, are controlled to flow within the double phases.

Based on the above-mentioned discussion, the detailed distribution of the voltages and currents in the proposed DC-DC MMC is depicted in Figure 3. As depicted, the red dashed lines denote the current flow routes of the DC components in the proposed DC-DC MMC. The blue dashed line represents the current flow route of the alternative current. It is assumed that the left and right phases in the proposed DC-DC MMC have symmetrical parameters. Therefore, the input DC current is split evenly for the left and right upper arms. As for the DC components of the left and right lower arm currents, they are directly controlled to flow into the output side, constituting the output current with the DC components of the left and right upper arm currents.

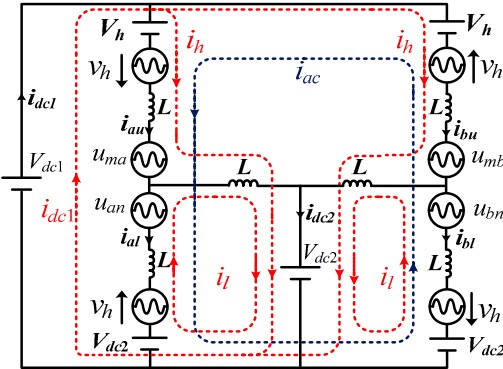

**Figure 3.** The equivalent model of the proposed DC-DC MMC.

Accordingly, the preferred arm voltage references for the proposed DC-DC MMC are given as follows:

$$\begin{cases} u_{au} = V_h - V_{ac}\sin\omega t \\ u_{al} = V_{dc2} + V_{ac}\sin\omega t \\ u_{bu} = V_h + V_{ac}\sin\omega t \\ u_{bl} = V_{dc2} - V_{ac}\sin\omega t \end{cases} \tag{3}$$

where $u_{au}$ and $u_{al}$ are the upper and lower arm voltage references of phase *a* and $u_{bu}$ and $u_{bl}$ are the upper and lower arm voltage references of phase *b*. Furthermore, $V_h$ is the DC component of the upper arm voltage reference, $V_{dc2}$ is the DC component of the lower arm voltage reference, and $V_{ac}$ and $\omega$ are the amplitude and the angular frequency of the inserted AC voltage $v_h$, respectively.

According to the current flow routes in Figure 3, the upper and lower arm currents of the double phases can be expressed as

$$
\begin{cases}
i_{au} = i_h + i_{ac} \\
i_{al} = -i_l + i_{ac} \\
i_{bu} = i_h - i_{ac} \\
i_{bl} = -i_l - i_{ac}
\end{cases}
\tag{4}
$$

where $i_{au}$ and $i_{al}$ are the upper and lower arm currents for phase $a$, $i_{bu}$ and $i_{bl}$ are the upper and lower arm currents for phase $b$, and $i_h$ and $i_l$ are the DC components of the upper and lower arm currents for each phase, respectively. Moreover, $i_{ac}$ is the injected alternative current (also called the circulating current), which is defined as

$$
i_{ac} = I_{ac} \sin \omega t
\tag{5}
$$

where $I_{ac}$ is the amplitude of the circulating current. Based on the current distribution in Figure 3, the output current $i_{dc2}$ can be expressed as

$$
i_{dc2} = 2i_h + 2i_l
\tag{6}
$$

Equation (6) indicates that the output current is contributed by the DC components of all of the arm current in the proposed DC-DC MMC. Additionally, it also can be demonstrated that the output current is split evenly between the left and right phases. Therefore, referring to the route of the current flow highlighted in Figure 3 and Equation (6), the input current of the proposed DC-DC MMC can be expressed as

$$
i_{dc1} = 2i_h
\tag{7}
$$

where $i_{dc1}$ denotes the input DC current. Based on Equations (6) and (7), it can be found that the input DC current is entirely delivered to the output side. Therefore, the step-up conversion between the input current and output current is mainly achieved by adjusting the DC components $i_l$ of the lower arm currents.

### 2.3. Power Analysis of the Proposed DC-DC MMC

The cascaded HBSMs and the middle cells in the proposed DC-DC MMC occupy the vital positions for distributing the system's powers. For simplification, phase $a$ is taken into consideration in the discussion below. Considering Equations (3)–(7), the upper arm and the lower arm powers for each phase can be expressed as

$$
p_{au} = V_h i_h - \frac{1}{2} V_{ac} I_{ac} - V_{ac} i_h \sin \omega t + V_h I_{ac} \sin \omega t + \frac{1}{2} V_{ac} I_{ac} \cos 2\omega t
\tag{8}
$$

$$
p_{al} = -V_{dc2} i_l + \frac{1}{2} V_{ac} I_{ac} - V_{ac} i_l \sin \omega t + V_{dc2} I_{ac} \sin \omega t - \frac{1}{2} V_{ac} I_{ac} \cos 2\omega t
\tag{9}
$$

where $p_{au}$ and $p_{al}$ are the upper and lower arm powers of phase $a$, respectively. The voltage conversion ratio of the DC-DC MMC is defined as

$$
r = \frac{V_{dc2}}{V_{dc1}} = \frac{i_{dc1}}{i_{dc2}}
\tag{10}
$$

where $r$ is the voltage conversion ratio of the DC-DC MMC. It was concluded in [21] that the balanced state of the HBSMs could only be satisfied when the active power consumptions of the HBSMs remained zero. Therefore, based on Equations (4) and (6)–(9), this yields the following:

$$
\frac{V_h}{V_{dc2}} = \frac{V_{dc1} - V_{dc2}}{V_{dc2}} = \frac{1}{r} - 1 = \frac{i_{dc2} - i_{dc1}}{i_{dc1}} = \frac{i_l}{i_h}
\tag{11}
$$

Furthermore, considering Equation (11), it is concluded that

$$V_h i_h = V_{dc2} i_l \tag{12}$$

Referring to [21], the common mode and differential mode powers of the proposed DC-DC MMC can be expressed as

$$\begin{cases} p_{au} = p_{acom} + p_{adif} \\ p_{al} = p_{acom} - p_{adif} \end{cases} \tag{13}$$

where

$$\begin{cases} p_{acom} = -\frac{i_h + i_l}{2} V_{ac} \sin \omega t + \frac{1}{2} V_{dc1} I_{ac} \sin \omega t \\ p_{adif} \quad = -\frac{i_h - i_l}{2} V_{ac} \sin \omega t \\ \qquad + \frac{V_h - V_{dc2}}{2} I_{ac} \sin \omega t + \frac{1}{2} V_{ac} I_{ac} \cos 2\omega t \end{cases} \tag{14}$$

where $p_{acom}$ and $p_{adif}$ are the common mode and differential mode powers of $p_{au}$ and $p_{al}$, respectively. In Equation (14), the common mode power in $p_{acom}$ contains the fundamental frequency component, while the differential mode power in $p_{adif}$ contains both the fundamental and second-order harmonic components. Considering different $r$ values, the upper and lower arm powers contain different values of the fundamental and second-order frequency components. In particular, it can be observed from Equation (14) that the fundamental frequency part in $p_{adif}$ can be entirely eliminated when $r = 0.5$. Thus, the fundamental frequency powers of the upper and lower arms in the DC-DC MMC are mainly contributed by the common mode powers in Equation (14) when $r = 0.5$. This implies that the upper and lower arm powers of each phase contain the same fundamental frequency component. However, for the second-order harmonic part in $p_{adif}$ posing a differential feature, this always results in the same amplitude with the opposite sign of the second-order harmonics in the upper and lower arm powers.

According to [21], it has been verified that the amplitude of the $i$th frequency component in the SM capacitor voltage ripples shows a specific proportion to its corresponding frequency parts in the arm powers. Based on Equation (13), the HBSM capacitor voltages of the proposed MMC can be calculated as follows:

$$\begin{cases} u_{acmu} = u_{acom} + u_{adif} \\ u_{acml} = u_{acom} - u_{adif} \end{cases} \tag{15}$$

where

$$\begin{cases} u_{acom} = \frac{i_h + i_l}{2} \frac{V_{ac}}{Cn\omega U_{ref}} \cos \omega t - \frac{V_{dc1} I_{ac}}{2Cn\omega U_{ref}} \sin \omega t \\ u_{adif} \quad = \frac{i_h - i_l}{2} \frac{V_{ac}}{Cn\omega U_{ref}} \cos \omega t \\ \qquad - \frac{V_h - V_{dc2}}{2} \frac{I_{ac}}{Cn\omega U_{ref}} \cos \omega t + \frac{V_{ac} I_{ac}}{4Cn\omega U_{ref}} \sin 2\omega t \end{cases} \tag{16}$$

where $u_{acmu}$ and $u_{acml}$ express the HBSM capacitor voltages of the upper and lower arms in the proposed DC-DC MMC, respectively, and $u_{acom}$ and $u_{adif}$ are the common mode voltage and differential mode voltage of $u_{acmu}$ and $u_{acml}$, respectively. Here, $C$ is the HBSM capacitance, $n$ is defined as the HBSM number in each arm, and $U_{ref}$ denotes the HBSM capacitor voltage reference. From Equation (16), it is determined that the HBSM capacitor voltage ripples mainly contain the fundamental and second-order harmonic components in the DC-DC MMC. The fundamental frequency parts of the HBSM capacitor voltage ripples in both the upper and lower arms occupy a dominant position. Therefore, different from the voltage shape of the HBSM capacitor voltage ripples under a rectifier or inverter MMC, the HBSM capacitor voltage ripples of the upper and lower arms in the proposed DC-DC MMC have nearly the same shape.

### 2.4. Analysis of the Inserted Alternative Voltage and Current

According to the energy conservation law and Equations (6) and (7), the input and output powers can be expressed in the following form:

$$V_{dc1} \cdot i_{dc1} = V_{dc1} \cdot 2i_h = V_{dc2} \cdot i_{dc2} = V_{dc2}(2i_h + 2i_l) \tag{17}$$

Referring to Equations (11), (12), and (17), this yields

$$V_{dc2}i_l = \frac{V_{dc1} \cdot 2i_h - V_{dc2} \cdot 2i_h}{2} = \frac{1 - (V_{dc2}/V_{dc1})}{2}V_{dc1} \cdot 2i_h = \frac{1}{2}(1-r)P \tag{18}$$

where

$$P = V_{dc1}i_{dc1} = V_{dc2}i_{dc2} \tag{19}$$

where $P$ is defined as the rated power of the proposed DC-DC MMC. Once the HBSMs of the proposed DC-DC MMC are balanced appropriately, this also demonstrates that the DC components of the upper and lower arm powers in Equations (8) and (9) remain zero, which yields

$$V_h i_h = \frac{1}{2}V_{ac}I_{ac} = V_{dc2}i_l \tag{20}$$

Therefore, based on Equations (18) and (20), it is concluded that

$$V_{ac}I_{ac} = 2V_{dc2}i_l = (1-r)P \tag{21}$$

The term $V_{ac}I_{ac}$ on the left side of Equation (21) expresses the required injected alternative power for supporting the balanced state of the HBSM capacitor voltages. It also demonstrates that the term $V_{ac}I_{ac}$ can be defined once the system parameters of the proposed DC-DC MMC are specified. In other words, $V_{ac}I_{ac}$ should be equal to $(1-r)P$ in order to satisfy the balanced requirement of the proposed DC-DC MMC. Consequently, an alternative form of Equation (21) can be expressed as

$$I_{ac} = \frac{1-r}{V_{ac}}P \tag{22}$$

Equation (22) implies that the amplitude of the injected alternative current is inversely proportional to the amplitude of the inserted alternative voltage once the voltage conversion ratio and the rated power are defined. Therefore, for the purpose of reducing the power loss and current stress of the power devices, the decreased amplitude of the injected current is preferred by increasing the amplitude of the inserted alternative voltage.

However, the maximum value of the injected alternative voltage shows a close relation with the HBSM features. In the proposed DC-DC MMC, the upper and lower arm voltage references consist of the DC and alternate voltage components. Since the cascaded HBSMs could only output positive voltages, the maximum value of the injected alternative voltage is restricted by the DC component of the HBSM voltage reference obtained from Equation (3). Furthermore, the DC components of the upper and lower arm voltage references are not the same during most cases, which implies that the maximum amplitude value of the injected alternative voltage is determined by the smaller DC component between the upper and lower arm voltage references. Consequently, the maximum amplitude value of the injected alternative voltage is equal to half of the smaller one between $V_h$ and $V_{dc2}$. This specifically leads to the conclusion that the maximum amplitude value of the alternative voltage can be obtained when $r = 0.5$, which simultaneously proposes the minimum amplitude value of the alternative current once $(1-r)P$ is unchanged. In other cases, the maximum value of $V_{ac}$ is restrained by $V_h$ and $V_{dc2}$, which obeys the following rule:

$$V_{ac} \leq \min(0.5V_h, \ 0.5V_{dc2}) \tag{23}$$

Equation (23) shows that a smaller voltage conversion ratio $r$ induces a bigger difference between $0.5V_h$ and $0.5V_{dc2}$. Under this solution, a decreased amplitude of $V_{ac}$ should be preferred, which accompanies the increased amplitude of $i_{ac}$ with additional power losses.

### 2.5. Analysis of the Middle Cells

The middle cells in the proposed DC-DC MCC serve as the active power filter. Based on the discussion in Equation (3), the undesired harmonic voltages would be imposed at the output side due to the inverse feature of the injected alternative voltages in the upper and lower arms. However, the adverse influence can be suppressed significantly by having suitable control of the middle cells to eliminate undesirable harmonics. Unlike the MMC topologies used to filter out the undesirable voltages with bulky passive LC components [16], the middle cells in the proposed DC-DC MMC can produce any required voltages accurately. It can conveniently adjust the amplitude and the angular frequency of the required alternative voltage by modifying the voltage references of the middle cells. Consequently, considering the direction marked for the voltage reference in Figure 3, the references of the middle cells can be concluded to be

$$\begin{cases} u_{ams} = V_{ac} \sin \omega t \\ u_{asn} = -V_{ac} \sin \omega t \\ u_{bms} = -V_{ac} \sin \omega t \\ u_{bsn} = V_{ac} \sin \omega t \end{cases} \tag{24}$$

where $u_{ams}$ is the terminal voltage between $m$ and $s$ in the middle cell of phase $a$, $u_{asn}$ is the terminal voltage between $s$ and $n$ in the middle cell of phase $a$, $u_{bms}$ is the terminal voltage between $m$ and $s$ in the middle cell of phase $b$, and $u_{bsn}$ is the terminal voltage between $s$ and $n$ in the middle cell of phase $b$. Based on the definition in Equation (24), the alternative voltage component can be produced only when the terminals can output both the positive and negative SM capacitor voltages. For simplification, the terminal voltage $u_{ams}$ was taken as an example for analysis. Thus, referring to the basic switching states listed in Table 1, the terminal voltage $u_{ams}$ can be composed of either the former two switching states in Table 1 or the fifth and sixth switching states in Table 1. However, when the fifth and sixth switching states in Table 1 are adopted to produce $u_{ams}$, an unexpected alternative voltage is imposed on the terminal voltage $u_{amn}$. Under this condition, the terminal voltage $u_{amn}$ would be inserted into the phase voltage, which disturbs the arm voltage distribution. Therefore, the former two switching states in Table 1 are adopted to control the middle cell due to the reason that the terminal voltage $u_{amn}$ always remains zero. According to the above-mentioned discussion, similar analysis results can be applied for other terminal voltages in Equation (24). Consequently, the former two switching states in Table 1 are preferred.

Figure 4 illustrates the equivalent single-phase structure of the proposed DC-DC MMC. This can be considered the buck converter, while the middle cell acts as the low-pass filter. In terms of the single-phase converter, since there is no additional current flow loop, the injected alternative current will inevitably invade the input DC current. This enables the superior feature of the double-phase structure in the proposed DC-DC MMC, which supplies an active current flow loop for the injected alternative current.

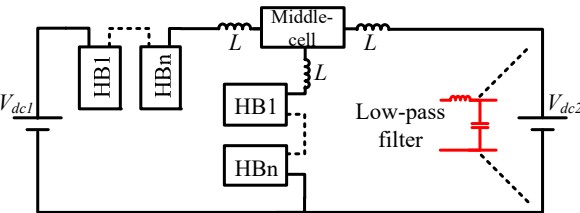

**Figure 4.** Single-phase DC-DC MMC in buck converter form.

To further investigate the structure feature of the proposed middle cell, the distributions of the voltages and currents of the middle cell should be studied. Referring to the marked currents of the middle cell in Figure 2b, based on Equations (3)–(7), the following currents can be derived:

$$
\begin{cases}
i_{amidm} = -\frac{1}{2}\left(1 + \frac{V_{ac}}{U_{mid}}\sin\omega_h t\right)(i_h + i_{ac}) \\
i_{amidn} = \frac{1}{2}\left(1 + \frac{V_{ac}}{U_{mid}}\sin\omega_h t\right)(-i_l + i_{ac}) \\
i_{amids} = \frac{1}{2}\left(1 - \frac{V_{ac}}{U_{mid}}\sin\omega_h t\right)(i_h + i_l)
\end{cases}
\tag{25}
$$

where $i_{amidm}$ is the switching current of $s_{x1}$ of phase $a$, $i_{amidn}$ is the switching current of $s_{x3}$ of phase $a$, $i_{amids}$ is the switching current of $s_{x5}$ of phase $a$, and $U_{mid}$ is defined as the middle cell capacitor voltage reference. Equation (25) indicates that the DC components of $i_{amidm}$ and $i_{amidn}$ are directly transferred to the output DC side, which imposes no influence on the balanced state of the middle cell capacitor voltage. Consequently, referring to Equation (25), the middle cell capacitor current can be concluded to be

$$
i_{amid} = -(i_{midmu} + i_{midnu} + i_{midxu}) == \frac{V_{ac}}{U_{mid}}(i_h + i_l)\sin\omega t
\tag{26}
$$

From Equation (26), it is demonstrated that the middle cell capacitor current only contains the fundamental frequency component. Consequently, the middle cell capacitor voltage can be derived as follows:

$$
u_{amid} = U_{mid} - \frac{V_{ac}(i_h + i_l)}{C_{mid}\omega U_{mid}}\cos\omega t
\tag{27}
$$

where $u_{amid}$ is the middle cell capacitor voltage of phase $a$ and $C_{mid}$ is defined as the middle cell capacitance. From Equation (27), it can be concluded that the middle cell capacitor voltage ripple is directly proportional to the load current of the proposed DC-DC MMC.

### 2.6. Comparison of the Proposed Converter and Existing DC-DC MMC

A feature comparison of the push–pull M2DC [16], FC-MMC [18], AM-MMC [20], and the proposed converter is provided. The comparison items include the SM number, IGBT number, bulky capacitor, bulky inductor, current stress, and reliability. As for the detailed comparison parts listed in Table 2, obviously, among the listed MMC topologies, although the proposed converter has a larger IGBT number, this is not required to equip the bulky LC filter components. Additionally, the proposed converter has less current stress among the listed topologies. Meanwhile, in terms of the reliability, since the middle cells in the proposed converter serve as the active power filters, evenly distributed losses can be achieved among the IGBTs, which improves the reliability of the middle cells.

**Table 2.** Comparison of the existing DC-DC MMC when $r = 0.5$.

|  | Push-Pull M2DC | FC-MMC | AM-MMC | Proposed |
|---|---|---|---|---|
| SM number | $4n$ | $4n$ | $6n$ | $4n + 2$ |
| IGBT number | $12n$ | $8n$ | $16n$ | $14n$ |
| Bulky capacitor | Not required | Required | Not required | Not required |
| Bulky inductor | Required | Not required | Not required | Not required |
| Current stress | High | High | High | Low |
| Reliability | High | Low | High | High |

## 3. Control Structure of the Proposed DC-DC MMC

Since the terminal voltages $u_{mn}$ of the middle cells always remain zero when adopting the former two switching states in Table 1 during normal operation of the proposed DC-DC MMC, the balancing control of the HBSMs has a similar feature to that of the traditional

MMC [9]. Considering the above-mentioned discussion related to Figure 3, the HBSMs are in charge of achieving active power transmission, while the middle cells are responsible for eliminating undesired harmonics. Therefore, the average value of all HBSM capacitor voltages of the single phase should be controlled to be zero. The control of the proposed DC-DC MMC consists of the average control and the balancing control. The structure of the average control is shown in Figure 5. Based on Equation (14), the fundamental components contribute the main parts in $p_{com}$. As discussed in Section 2, the upper and the lower arm HBSM capacitor voltages contain the same fundamental frequency component. Therefore, $\Delta u_{cm}$ in Figure 5 mainly contains the DC and fundamental frequency components. This arrangement is entirely different from that of the DC/AC MMC in [22] that consists of the DC and second-order components. Due to this, the alternate circulating current in the proposed DC-DC MMC mainly contains the fundamental part, whereas the DC/AC MMC contains the second-order part. Then, $\Delta u_{cm}$ is applied to the proportional and integral (PI) controller in order to obtain the inner loop current reference. The inner loop current reference contributes to the common part of both the upper and lower arm currents. It should be noted that the inner loop current reference $i_{dref}$ in Figure 5 contains the DC part and the circulating current, which can be expressed as

$$i_{xref} = \frac{i_h - i_l}{2} \pm i_{ac} \tag{28}$$

where $i_{xref}$ is the inner loop current reference for phase $x$ ($x = a$, $b$) and the symbol $\pm$ differentiates the left and the right phases. Equation (28) indicates that the inner loop current reference consists of the common part $(i_h - i_l)/2$ and the differential part $i_{ac}$. The common part in Equation (28) flows from the input to the output sides, contributing to the output DC current, while the differential current in Equation (27) flows only within the double phases without disturbing the output current. Finally, the compensation part of the average control $\Delta v_x$ ($x = a$, $b$), being responsible for the balance states between the upper and lower arms, is sent to the modulation process.

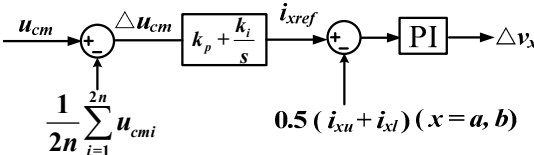

**Figure 5.** Average control of the proposed DC-DC MMC.

For the individual control, closed-loop control with a proportional controller is proposed as illustrated in Figure 6. Figure 6a depicts the balancing control of the HBSM, while Figure 6b shows the balancing control of the middle cell. In Figure 6a, the compensation part of the HBSM balancing control can be obtained by the product of the output of the P controller and the polarity of the corresponding arm current. Typically, the active power is absorbed from the arm currents to maintain the balanced state of the HBSM capacitor voltages. In Figure 6b, unlike the balancing control of the HBSM, the active power, required to maintain the balanced state of the middle cell capacitor voltage, is absorbed from the output current. The purpose of the individual control in Figure 6 is to prevent the SM capacitor voltage from deviating from its reference, which is mainly achieved by producing a corresponding compensation part in the modulation waveforms. Consequently, the voltage reference of the HBSM can be described as follows:

$$\begin{cases} u_{ausmi} = \frac{V_h}{n} - \frac{v_h}{n} - \Delta v_a + \Delta u_{ausmi} \\ u_{alsmi} = \frac{V_{dc2}}{n} + \frac{v_h}{n} - \Delta v_a + \Delta u_{alsmi} \\ u_{busmi} = \frac{V_h}{n} + \frac{v_h}{n} - \Delta v_b + \Delta u_{busmi} \\ u_{blsmi} = \frac{V_{dc2}}{n} - \frac{v_h}{n} - \Delta v_b + \Delta u_{blsmi} \end{cases} \tag{29}$$

where $u_{xusmi}$ and $u_{xlsmi}$ are the *i*th voltage references for the upper and the lower arms of phase $x$ ($x = a, b$), respectively.

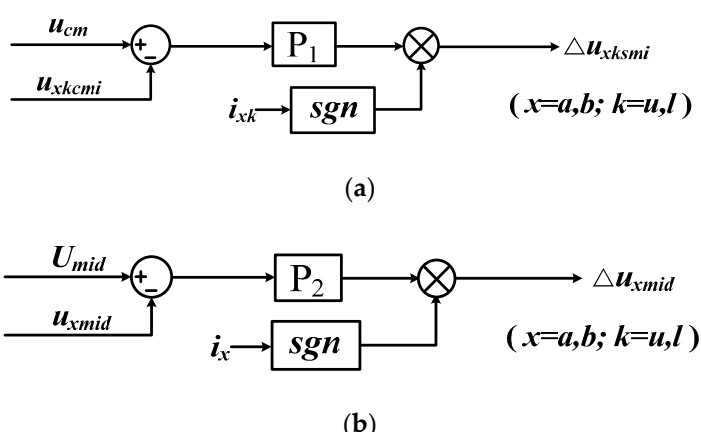

(a)

(b)

**Figure 6.** Individual balancing control of SM capacitor voltage. (**a**) The balancing control of the HBSM capacitor voltage. (**b**) The balancing control of the middle cell capacitor voltage.

Considering the voltage reference of the middle cell, due to the complementary features of $u_{mx}$ and $u_{xn}$, it is sufficient to balance the middle cell capacitor voltage for $u_{mx}$ only. Thus, it is concluded that

$$\begin{cases} u_{amidref} = v_h + \Delta u_{amid} \\ u_{bmidref} = -v_h + \Delta u_{bmid} \end{cases} \quad (30)$$

where $u_{xmidref}$ is the middle cell voltage reference of phase $x$ ($x = a, b$). Referring to the former two switching states in Table 1, once the references in Equation (30) are defined, the switching states of the middle cell can be defined.

## 4. Simulation Results

The theoretical analysis and control strategy of the proposed DC-DC MMC were comprehensively discussed in Sections 2 and 3. In order to evaluate and verify the performance of the proposed DC-DC MMC, simulation results using the MATLAB platform are presented in this section. As per the simulation parameters listed in Table 3, each of the arms in the proposed DC-DC MMC consisted of eight HBSMs. The HBSM capacitor voltage reference was considered to be 500 V, which was aimed to support the input DC bus voltage (4000 V). In order to produce the required compensation parts from the middle cells, the middle cell voltage reference was set to 2000 V. Both of the switching frequencies of the HBSM and middle cell were set to be the same (1 kHz). Moreover, considering the superior output performance of the cascaded HBSMs, a phase-shifted carrier PWM (PSC-PWM) was adopted for modulation process.

**Table 3.** Circuit parameters for simulation.

| Symbol | Quantity | Value |
|--------|----------|-------|
| $V_{dc}$ | DC link voltage | 4000 V |
| $N$ | Number of SMs per arm | 8 |
| $C$ | HBSM capacitor | 1 mF |
| $C_{mid}$ | FBSM capacitor | 1 mF |
| $L$ | Arm inductor | 2 mH |
| $f_c$ | SM switching frequency | 1 kHz |
| $f_{fc}$ | Middle cell switching frequency | 1 kHz |
| $f_h$ | Frequency of inserted AC voltage | 100 Hz |
| $u_{cm}$ | HBSM capacitor voltage reference | 500 V |
| $U_{mid}$ | Middle cell voltage reference | 2000 V |
| $R_{load}$ | Load resistance | 100 Ω |

Figure 7 illustrates the simulation results of the upper and lower arm voltages. Based on the control strategy analyzed in Section 3, the output voltages of the upper and lower arms contained the dominant DC and alternative components. The difference lied in that the alternative voltages of the upper and lower arms had a phase shift of π.

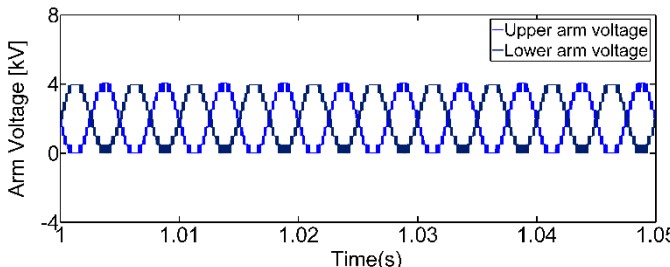

**Figure 7.** The upper arm and lower arm voltages in the proposed DC-DC MMC.

Figure 8 depicts the simulation results for the HBSM capacitor voltages in the upper and lower arms when $r = 0.5$. As discussed in Section 2, the fundamental frequency component of the differential mode voltage in Equation (16) could be completely eliminated when $r = 0.5$. Therefore, the HBSM capacitor voltage ripples mainly contained the fundamental frequency components. In addition, it can be observed that the upper and lower HBSM capacitor voltage ripples had a similar shape. As for the slight difference between the two waveforms in Figure 8, this was mainly caused by the second-order harmonic components in the differential mode voltage in Equation (16). The simulation result of the capacitor voltage of the middle cell is illustrated in Figure 9. Referring to the discussion in Equation (26), the capacitor voltage ripple of the middle cell mainly contained the fundamental frequency component, which showed accordance with the waveform in Figure 9.

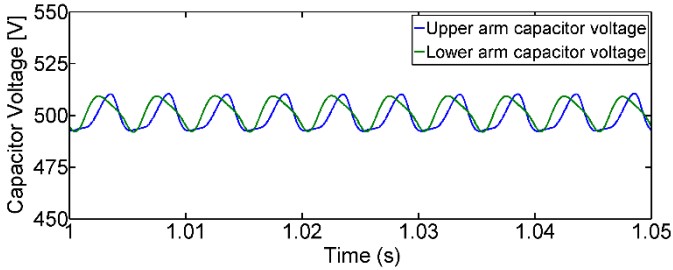

**Figure 8.** The upper arm and lower arm capacitor voltages of the proposed DC-DC MMC when $r = 0.5$.

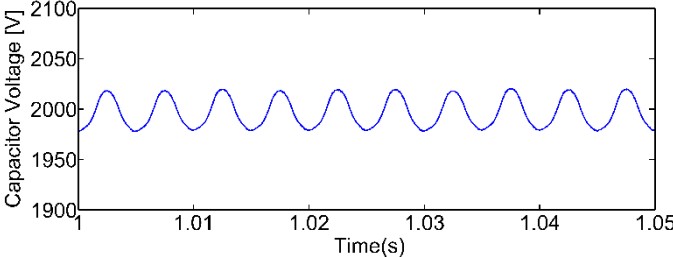

**Figure 9.** The capacitor voltage of the middle cell for the proposed DC-DC MMC.

Figure 10 depicts the upper and lower arm currents when $r = 0.5$. As discussed in Section 2, the upper and lower arm currents in the single phase contained the same alternative current, while the difference lied in that they had distinct DC components. Ultimately, the DC components of the upper and lower arm currents were controlled to flow from the arms to the output side, forming the load current.

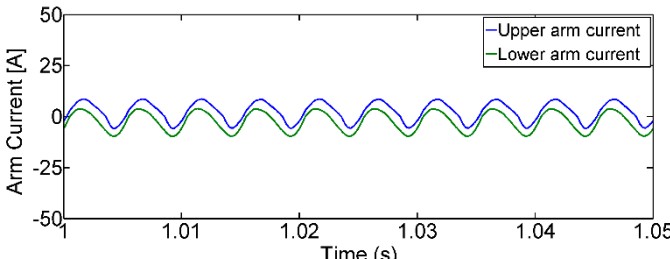

**Figure 10.** The upper arm and lower arm currents of the proposed DC-DC MMC when *r* = 0.5.

In order to verify the effectiveness of the proposed DC-DC MMC, the simulation results under the DC-DC MMC without the middle cells were also found. Figure 11 illustrates the comparison of results for the output voltages when *r* = 0.5. Figure 11a shows the simulation results of the output voltage for the DC-DC MMC without adopting the middle cells, while Figure 11b shows the simulation results of the output voltage in the proposed DC-DC MMC. As we can see from Figure 11, a serious ripple for the output voltage mainly caused by the injected alternative voltages can be observed. In addition, to evaluate the dynamic performance of the proposed DC-DC MMC, step change analysis of the output voltage was also performed. Figure 12 shows that a fast response from the new steady state for the output voltage could be observed when the reference of the output voltage was increased from 1700 V to 2300 V. Under the same conditions, the step change of the upper and lower arm currents is also provided in Figure 13, which verifies the fast response of the arm current for the proposed DC-DC MMC.

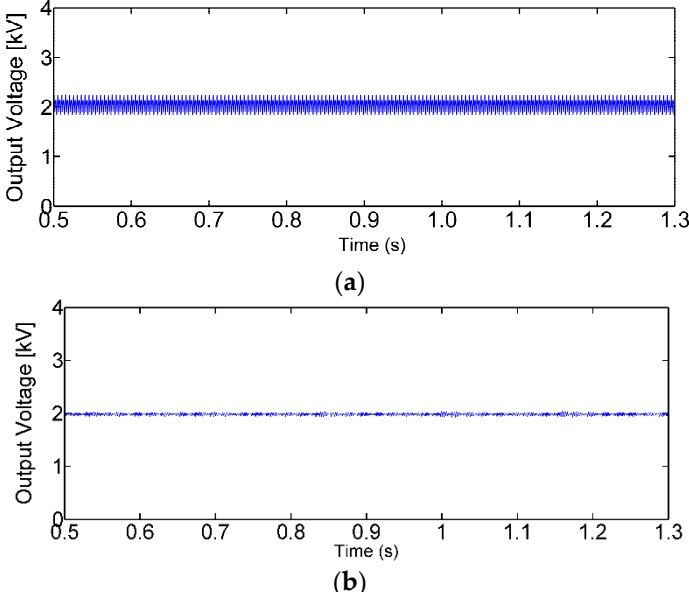

**Figure 11.** Comparison of the output voltages for different DC-DC MMC topologies when *r* = 0.5. (**a**) Output voltage of the DC-DC MMC without adopting the middle cells. (**b**) Output voltage of the proposed DC-DC MMC.

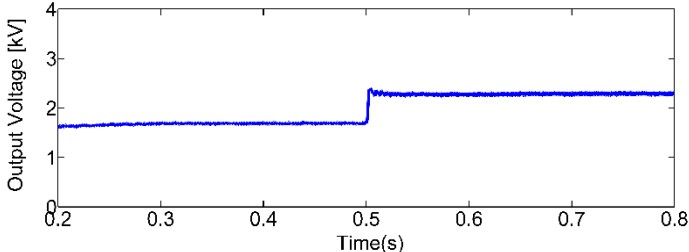

**Figure 12.** The DC output voltage with step change from 1.7 kV to 2.3 kV in the proposed DC-DC MMC.

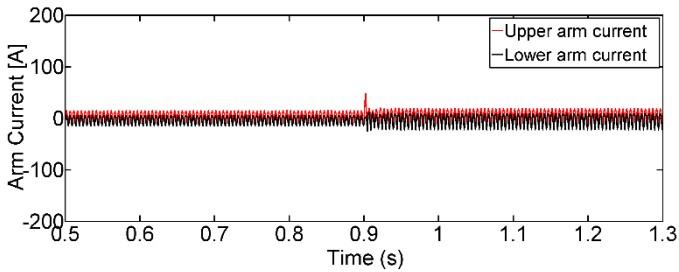

**Figure 13.** The upper and lower arm currents when voltage step changes from 1.7 kV to 2.3 kV in the proposed DC-DC MMC.

## 5. Experimental Results

To verify the effectiveness of the theoretical analysis of the proposed DC-DC MMC, a power prototype was built. It consisted of the power circuit and the control system. Due to the restrictive experimental facility, the power stage of the proposed DC-DC MMC was developed using four HBSMs per phase leg, and each phase was equipped with one middle cell, as shown in Figure 14. Meanwhile, a three-phase uncontrolled rectifier was employed to produce the required DC input voltage. The detailed structure of the control system is presented in Figure 15, where the DSP and FPGA were the main components. The DSP was developed with the help of the TMS320f28335 chip to achieve the control algorithm and to produce the modulation waveforms for the FPGA. The FPGA system was based on the EP4CE115F23C8N chip to achieve analog sampling and PWM signal collection. The Altera Cyclone IV EP4CE115F23C8N devices were targeted toward high-volume, cost-sensitive applications, enabling system designers to meet increasing bandwidth requirements while lowering costs. It covered 110,000 logic elements, 4 Mb embedded memory, 4 general-purpose PLLs, and 528 maximum user I/Os. Therefore, it was proper to produce the large number of PWM-driven signals and the analog sampling signals. During normal operation, the DSP chip acted as the host platform, while the FPGA system served as the affiliated platform. An experimental photograph of the power prototype for the proposed DC-DC MMC is shown in Figure 16, while the experimental parameters of the proposed DC-DC MMC are listed in Table 4.

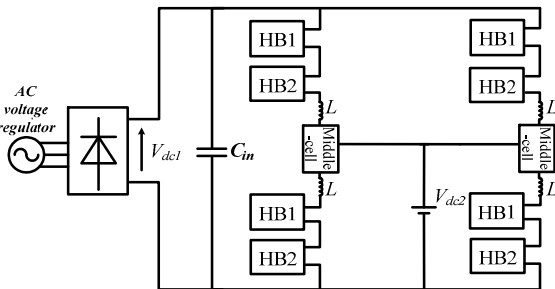

**Figure 14.** Power circuit of the proposed DC-DC MMC.

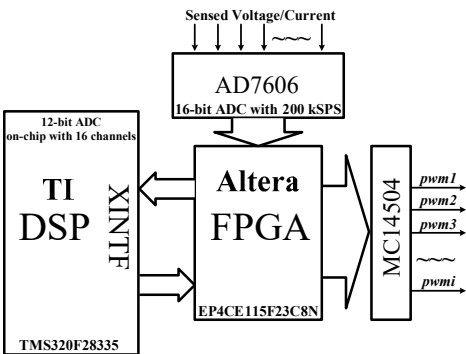

**Figure 15.** Diagram of the control system.

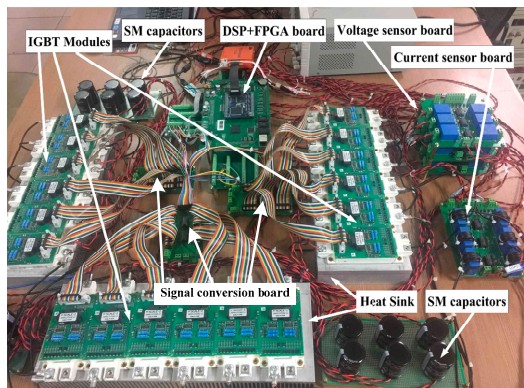

**Figure 16.** Experimental photograph of the proposed DC-DC MMC.

**Table 4.** Circuit parameters for experiment.

| Symbol | Quantity | Value |
|--------|----------|-------|
| $V_{dc}$ | DC link voltage | 200 V |
| $N$ | Number of SMs per arm | 2 |
| $C$ | SM capacitor | 0.68 mF |
| $C_{mid}$ | Middle cell capacitor | 1 mF |
| $L$ | Arm inductor | 2 mH |
| $f_c$ | SM switching frequency | 1 kHz |
| $f_{fc}$ | Middle cell switching frequency | 1 kHz |
| $f_h$ | Frequency of inserted voltage | 50 Hz |
| $u_{cm}$ | SM capacitor voltage reference | 100 V |
| $U_{mid}$ | Middle cell capacitor voltage reference | 100 V |
| $R_{load}$ | Load resistance | 25 $\Omega$ |

Figure 17 shows the experimental results of the input current and the middle cell capacitor voltage at $r = 0.4$. In the proposed converter, the injected alternative current was controlled to keep its flow within the phase legs without disturbing the input current in Figure 17. The upper and lower arm currents in the single phase contained the same alternative current and the different DC parts, which indicates an agreement with the waveforms, as shown in Figure 18. The output voltage of the proposed converter at $r = 0.4$ is also presented in Figure 18, which offered a smooth feature. Finally, to verify the dynamic performance of the proposed DC-DC MMC, the step change from $r = 0.4$ to $r = 0.5$ (for the output voltage) is illustrated in Figure 19, which provided the fast response of the output voltage and arm current. In addition, the experimental results of the output voltage under the step change from $r = 0.4$ to $r = 0.6$ are also provided in Figure 20.

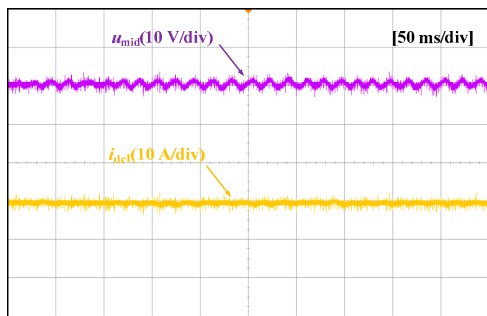

**Figure 17.** SM capacitor voltages of the proposed DC-DC MMC.

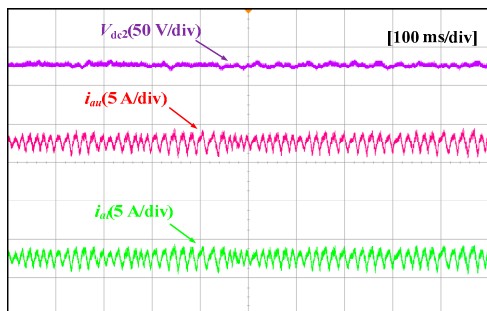

**Figure 18.** Output voltage and arm currents of the proposed DC-DC MMC.

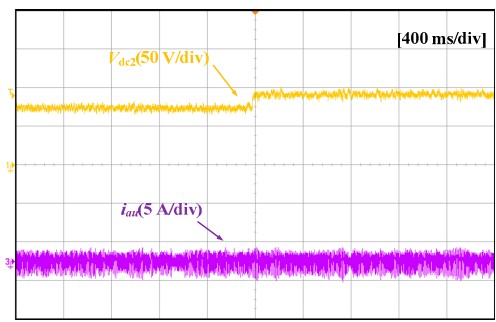

**Figure 19.** The DC output voltage and arm current of the proposed DC-DC MMC with a step voltage change from 80 V to 100 V.

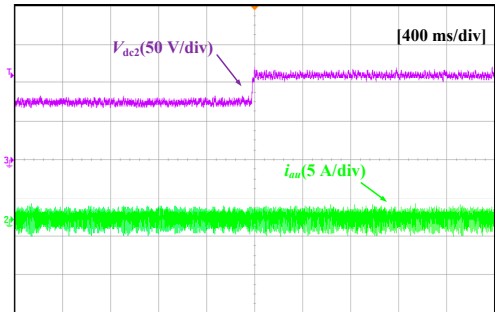

**Figure 20.** The DC output voltage and arm current of the proposed DC-DC MMC with a step voltage change from 80 V to 120 V.

## 6. Conclusions

This paper proposes a non-isolated double-phase DC-DC MMC while adopting novel middle cells. The main purpose of the proposed converter is to replace the bulky passive LC filer components with active ones. The specific procedure aims to prevent the injected balanced alternative harmonics from invading the input and output DC sides, which

is mainly achieved by the proposed circuit structure. During normal operation of the proposed DC-DC MMC, the alternative currents are controlled to flow within the double phase legs. This provides protection from any disruption to the input DC current. As for the injected alternative voltages, it was observed that the influence of the alternative voltages on the output DC side could be mostly suppressed by producing the composition parts from the middle cells. Consequently, the proposed middle cells serve as the active power filters for filtering the undesired harmonics on the output DC side in the DC-DC MMC. The relationship between the injected alternative voltage and current was comprehensively analyzed. It was concluded that the amplitude of the injected alternative current could be reduced by increasing the amplitude of the injected alternative voltage once the conversion ratio and load power were specified, and this procedure could be beneficial for reducing the power loss and switch current stress. Moreover, due to the specific structure of the middle cells, they showed no influence on the upper and lower arm voltages, providing a simple balance control for the SM capacitor voltages. The corresponding control strategy of the proposed converter was also discussed. Finally, the simulation and experimental results verified the performance of the proposed converter.

**Funding:** This article is supported by the Natural Science Basic Research Program of Shaanxi under grant 2021JQ-115, and in part by the Fundamental Research Funds for the Central Universities of China under Grant G2021KY0608.

**Institutional Review Board Statement:** Not applicable.

**Informed Consent Statement:** Not applicable.

**Conflicts of Interest:** The authors declare no conflict of interest.

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
