# Peer review of "A Non-Isolated DC-DC Modular Multilevel Converter with Proposed Middle Cells"

_electronics, doi:10.3390/electronics11071135_

Round 1

Reviewer 1 Report

This solution is based on the cost of introducing bulky LC filter components. For interconnecting different DC voltages in me dium voltage applications, this paper presents a non-isolated DC-DC MMC equipped with the proposed middle-cells. It aims to suppress the influence of the alternative voltages and currents at the input and output sides.

Experimentally it proposed that is theoretical analysis of the proposed DC-DC MMC, 463 a downscaled power prototype is built. (this doesn’t make any sense)

The results imply that it is not enough to analyse the attributes solely for the new information which is provided.

The main contribution, analyzing the potential effect is missing, also the amplitude of the injected alternative current which can be reduced by increasing the amplitude of the injected alternative voltage once the conversion ratio and load power are specified. Is not demonstrated properly.

Methods used in the study needs to be compared well.

A table of existing model/techniques must be added to to represent existing state of the art literature.

Conclusion section can be improved by highlighting key findings, limitation of the research and recommendation for future studies.

Add more standard and risk matrix in your study.

Reviewer 2 Report

The author has shown yet another type of MMC configuration and how it can be useful in the industry. Although the content is good the work can benefit from the following.

  1. Some minor corrections are required in written English.
  2. A mention of application areas/places where this technology can be used will add to the content. 
  3. when talking about the EP4CE115F23C8N chip, a brief into to this chip will be good.

Reviewer 3 Report

  1. The concerned literature corresponding to the DC-DC MMC are not well studied.
  2. What is the significance of Eq. (3) and Fig. 3? It is not well descried. Why are sinusoidal components introduced here? Which component is responsible for this and how?
  3. The DC-DC MMC lose its modularity feature with the suggested modifications while injecting the ac components. How you can clarify such situations.
  4. In Fig. 5 how you can decide the gains of the individual PI controller? Is there any such analysis for this?
  5. It was seen from Fig. 12 that the dynamic is not that much good while change in output voltage magnitude.
  6. Why the current ripple is going higher (in Fig. 13) when there is change in output voltage, clarify.
  7. More experimental analysis are required to evaluate the dynamic performance of the suggested modifications, such as change in input voltage, increase/decrease in output voltage.
  8. Analysis corresponding to any module failure would be much appreciated.
  9. Please analyze the gain factor of the suggested DC-DC MMC.
  10. Analyze how the current ripple of the suggested converter can be reduced.

Reviewer 4 Report

This paper presents a non-isolated DC-DC MMC equipped with the proposed middle-cells. It aims to suppress the influence of the alternative voltages and currents at the input and output sides. The abstract should be rewritten. Importance of the work should be pointed out through giving some specific results. In order to evaluate and verify the performance of the proposed DC-DC MMC, simulation results using MATLAB platform are presented in paper, but the simulation scheme is missing from the paper. The author should explain how he made the simulation scheme in MATLAB/SIMULINK and include it in the paper. In the conclusions in necessary to appear more emphasized the effects of different construction features upon the systems functioning.

Round 2

Reviewer 3 Report

Most of the comments are answered. Still, I have issue with the originality of the work. What would be application area for this topology.

Author Response

Please the attachment.
